# Iodine Status, Thyroid Function, and Birthweight: A Complex Relationship in High-Risk Pregnancies

**DOI:** 10.3390/jcm9010177

**Published:** 2020-01-08

**Authors:** Inés Velasco, Mar Sánchez-Gila, Sebastián Manzanares, Peter Taylor, Eduardo García-Fuentes

**Affiliations:** 1Pediatrics, Obstetrics and Gynecology Unit, Riotinto Hospital, Andalusian Health Service, Av. de la Esquila 5, 21660 Minas de Riotinto Huelva, Spain; 2Germans, Trias I Pujol Research Institute, Carretera de Canyet, s/n, Medicine Department, Autonomous University of Barcelona, 08916 Badalona, Spain; 3Department of Obstetrics & Gynecology, Virgen de las Nieves University Hospital, Andalusian Health Service, Av. de las Fuerzas Armadas 2, 18014 Granada, Spain; marsanchezgila@gmail.com (M.S.-G.); sebastian.manzanares.galan@gmail.com (S.M.); 4Thyroid Research Group, Systems Immunity Research Institute, Cardiff University School of Medicine, UHW, C2 Link Corridor, Heath Park, Cardiff CF14 4XN, UK; taylorpn@cardiff.ac.uk; 5Department of Gastroenterology, Virgen de la Victoria University Hospital, Biomedical Research Institute of Málaga-IBIMA, Biomedical Research Networking Center for Physiopathology of Obesity and Nutrition (CIBEROBN), 29010 Malaga, Spain; edugf1@gmail.com

**Keywords:** iodine, thyroid function, birthweight, amniotic fluid, high-risk pregnancy

## Abstract

(1) Background: The consequences of iodine deficiency and/or thyroid dysfunction during pregnancy have been extensively studied, emphasizing on infant neurodevelopment. However, the available information about the relationship between iodine, thyroid hormones, and fetal growth in high-risk pregnancies is limited. We aim to investigate if iodine metabolism and/or thyroid parameters can be affected by adverse antenatal/perinatal conditions. (2) Methods: A cross-sectional study examined differences in iodine status, thyroid function, and birthweight between high-risk (HR group; *n* = 108)) and low-risk pregnancies (LR group; *n* = 233) at the time of birth. Urinary iodine concentration (UIC), iodine levels in amniotic fluid, and thyroid parameters [thyroid-stimulating hormone (TSH), free thyroxine (FT4)] were measured in mother–baby pairs. (3) Results: There were significant differences between HR and LR groups, free thyroxine (FT4) concentration in cord blood was significantly higher in the LR group compared with HR pregnancies (17.06 pmol/L vs. 15.30 pmol/L, respectively; *p* < 0.001), meanwhile iodine concentration in amniotic fluid was significantly lower (13.11 µg/L vs. 19.65 µg/L, respectively; *p* < 0.001). (4) Conclusions: Our findings support the hypothesis that an adverse intrauterine environment can compromise the availability of FT4 in cord blood as well as the iodine metabolism in the fetus. These differences are more noticeable in preterm and/or small fetuses.

## 1. Introduction

Fetal growth is dependent upon a number of endocrine, paracrine, and autocrine events, involving interactions between the mother, placenta, and fetus, and these effects may program long-term physiology [1]. Thyroid hormones (TH) and their essential component, iodine, play a vital role in the early growth and development stages of most organs, especially the brain [2,3]. The relationship between iodine deficiency (ID) and fetal growth has been well established [4], and investigations of the impact of iodine and thyroid hormone transfer continue to improve our knowledge of maternal–fetal thyroid relationships [5].

Although marked physiological differences exist between the maternal and fetal thyroids, both systems interact through the placenta and the amniotic fluid, modulating the transfer of iodine and small but biologically important amounts of thyroid hormones from the mother to the fetus [6]. Iodine required for the fetal thyroid gland function comes from circulating iodine in the mother and deiodinization of iodothyronines in the placenta [7].

But many factors might hinder these physiological pathways, such as prematurity, severity of in utero compromise, maternal and perinatal comorbidities, complications associated with medically-induced preterm delivery. There is no exact definition of risk in pregnancy, but the tag “high-risk pregnancy” involves chronic health problems, such as diabetes or high blood pressure; infections; complications from a previous pregnancy; or other issues that might arise during pregnancy with an actual or potential hazard to the health or well-being of the mother or fetus [8]. All of them are likely to contribute to the vulnerability of the developing fetus and compromise its proper growth and maturation [9].

Small for gestational age (SGA) is used to describe those infants who are smaller in size than normal for their gestational age, defined as weight below the 10th percentile or 2 standard deviation (SD) for the gestational age [10]. The term intrauterine growth restriction (IUGR) is used to describe a fetus that cannot reach its growth potential due to placental insufficiency, and is associated with poor perinatal outcomes [11]. Although the two terms are different, SGA remains the best clinical surrogate for IUGR, since it would capture the majority of fetuses with this condition [12].

A recent systematic review has evaluated the association of maternal subclinical thyroid dysfunction and IUGR, demonstrating that subclinical hypothyroidism (SCH) exhibited a statistically significant association with IUGR [13]. Another systematic review has recently attempted to assess the effect of iodine repletion on human growth, but evidence of the impact of iodine supplementation, particularly on prenatal somatic growth, remains too uncertain [14].

Our group previously investigated the effect of physiological amounts of iodine intake on the iodine concentration in amniotic fluid [15] as well as the thyroid function and iodine status in healthy pregnant women at the time of birth [16]. The aim of the present study was: (i) to compare iodine status, thyroid function parameters, and birthweight between 2 groups of pregnant women classified as low-risk (LR group) and high-risk (HR group), respectively; (ii) to search for differences in maternal and neonatal thyroid function according to birthweight and/or gestational age; (iii) to analyze the effect of perinatal conditions (onset of labor, mode of delivery) on iodine metabolism and thyroid parameters at the time of birth.

## 2. Experimental Section

### 2.1. Study Subjects

This study used samples and data from the aforementioned study carried out at the Regional University Hospital, Málaga (Spain) [16]. These participants were considered as the low-risk group (LR group; *n* = 233), using as exclusion criteria any condition of obstetric and/or perinatal risk. Additionally, we recruited a new cohort of pregnant women who had been classified as high-risk pregnancies at the antenatal surveillance program of the Virgen de las Nieves University Hospital, Granada (Spain), from January 2015 to December 2018 (HR group; *n* = 108). Factors to be considered as high-risk pregnancies were maternal chronic diseases, preterm delivery, oligoamnios, growth restriction, or preeclampsia as well as history of adverse obstetric outcomes such as recurrent miscarriages or stillbirth. Inclusion criteria for HR group were pregnant women, previously classified at high-risk, admitted to hospital, and expected to give birth within the next 24 h. Criteria for exclusion included multiple gestation, fetal malformation, a known history of endocrine disease (thyroid dysfunction, diabetes), or exposure to iodinated agents (radiological contrast agents, iodized antiseptic solutions).

All participants from Málaga and Granada were pooled in a single database. In both groups, we analyzed maternal age, gestational age at delivery, and information regarding the consumption of supplements containing iodine during gestation, as well as timing and mode of delivery. The birthweight percentile was calculated according to customized weight curves for Spanish fetuses and newborns [17]. The study was approved by the Ethics and Clinical Investigation Committee of the Virgen de las Nieves University Hospital (Granada, Spain) and by the Ethics and Research Committee of Regional University Hospital (Málaga, Spain). Written informed consent was obtained from all the participants.

### 2.2. Laboratory Analysis

Blood samples and urine were collected from mothers during the active phase of delivery or just before elective caesarean section. A sample of amniotic fluid was obtained by aspiration, before amniorrhesis. At the time of birth, a sample of cord blood was obtained from the newborn and measurements were made of cord blood pH. All sample were stored at −80 °C and transferred to the laboratory of the Institute of Biomedical Research in Málaga (IBIMA) for analysis. Even though a considerable time lapse occurred between the first study [16] and the current one, all the laboratory procedures were replicated at the same laboratory, using exactly the same techniques and performed by the same biochemist.

The plasma levels of free thyroxine (FT4; pmol/L) and thyroid-stimulating hormone (TSH; mU/L) were measured by chemiluminescence immunoassay (Roche, Basel, Switzerland) at the clinical laboratory of the Regional University Hospital, Málaga. Urine and amniotic fluid iodine concentrations were measured following the Benotti technique and a modification for amniotic fluid samples [14,18]. The intra- and inter-assay coefficient of variation (CV) in urine samples were 3.2% and 3.5%, respectively, and in amniotic fluid they were 11.9% and 14.8%, respectively. The urinary iodine assay is subjected to a program of external quality assessment for the determination of iodine in urine by the Spanish Association of Neonatal Screening (AECNE) and by the Ensuring the Quality of Iodine Procedures (EQUIP) Program.

### 2.3. Statistical Analysis

Normal distribution of quantitative variables was assessed by the Kolmogorov–Smirnov test. Quantitative variables were shown as the mean ± standard deviation and qualitative variables as percentages. The contrast hypothesis for two samples was evaluated with the Student’s t-test, and for more than two samples, with an analysis of variance (ANOVA) test. The chi-square test was applied in case of categorical variables. The correlation between variables was determined using the Spearman test, designing multiple linear regression models in those cases where it was desired to predict the variance adjusted for other variables, besides the main variable. Logistic regression was applied in the case of dichotomous outcomes.

In all cases, the rejection level for a null hypothesis was alpha below 0.05. All data were analyzed using SPSS 20.0 (IBM SPSS Statistics, Armonk, NY, USA). 

## 3. Results

### 3.1. Clinical and Demographic Variables

All the maternal and neonatal variables included in the study are shown in Table 1. There were significant differences for all the clinical and analytical parameters, with the exception of Apgar scores and pH values in the umbilical cord: these neonatal variables were similar in low and high-risk groups. Apgar scores were considered as categorical variable (<5 and ≥5 at min 1; <7 and ≥7 at min 5) to describe the parameters precisely, but they also were treated as discrete quantitative variables in order to search for correlations with other variables.

No correlation was found between maternal age and gestational age at birth (*r* = −0.009, *p* = 0.876), nor between maternal age and birth weight (*r* = 0.001, *p* = 0.997), However, there were significant correlations between gestational age and birth weight (*r* = 0.588, *p* < 0.001), weight percentile (*r* = 0.201, *p* < 0.001), and 5-min Apgar score (*r* = 0.156, *p* = 0.004). Birth weight correlated with 1-min Apgar (*r* = 0.174, *p* = 0.001), and with 5-min Apgar (*r* = 0.438, *p* < 0.001). Similar but a bit weaker correlations were found between weight percentile and 1-min Apgar score (*r* = 0.122, *p* = 0.026), and with 5-min Apgar (*r* = 0.399, *p* = 0.001). None of the neonatal parameters (gestational age, birth weight, or weight percentile) correlated with pH values in umbilical cord (arterial nor venous).

Nevertheless, strong correlations were found between 1-min and 5-min Apgar scores (*r* = 0.577, *p* < 0.001), and between arterial and venous pH values in umbilical cord (*r* = 0.702, *p* < 0.001). There also were significant correlations between these four parameters: 1-min Apgar score correlated with arterial pH (*r* = 0.363, *p* < 0.001) and with venous pH in umbilical cord (*r* = 0.321, *p* < 0.001); and 5-min Apgar score correlated with arterial pH (*r* = 0.224, *p* = 0.004) and with venous pH in umbilical cord (*r* = 0.218, *p* = 0.001). These correlations between Apgar and pH values remained when we consider LR and HR pregnancies separately (Appendix A).

When preterm and at term pregnancies were compared, birth weight, weight percentile, and 5-min Apgar score were significantly lower in the preterm group (Appendix A). Besides that, when the comparison was made between the group of patients whose birth weight was under the 10th percentile and those with normal birth weight (percentiles 10–90), there also were significantly lower values in 1-min Apgar (8.51 ± 1.06 vs. 8.81 ± 0.66 respectively, *p* = 0.025) and 5-min Apgar scores (9.14 ± 0.71 vs. 9.75 ± 0.50 respectively, *p* < 0.001) and higher values in arterial pH (7.28 ± 0.07 vs. 7.25 ± 0.08 respectively, *p* = 0.049).

### 3.2. Iodine Status and Consumption of Supplements

The mean urinary iodine concentration (UIC) was significantly lower in the high-risk group compared to the low-risk group (Table 2). The median UIC was 126.86 and 103.38 µg/L for low-risk and high-risk group, respectively. The UIC did not correlate with maternal age (*r* = 0.058, *p* = 0.312), nor neonatal parameters such as gestational age (*r* = 0.009, *p* = 0.870), or birth weight (*r* = 0.018, *p* = 0.749). Almost a third of pregnant women (28.7%) of the high-risk group did not take iodine supplements (Table 1). Besides, 74.1% of HR women had a UIC <150 µg/L compared with 59.4% of LR women (*p* = 0.015). Maternal age was significantly lower in those with UIC <150 µg/L compared with pregnant women with UIC >150 µg/L (29.7 ± 5.9 years, vs. 31.0 ± 4.7 years, respectively; *p* = 0.033).

This difference in maternal age according to UIC was still present when we split the sample in LR and HR groups (29.1 ± 5.6 years, vs. 30.8 ± 4.7 years, for UIC < and UIC >150 μg/L respectively; *p* = 0.019 in the LR group and 30.9 ± 6.2 years, vs. 32.0 ± 4.4 years, for UIC < and UIC >150 μg/L respectively; *p* = 0.048 in the HR group). We did not find significant differences in gestational age, birth weight, or weight percentile according to UIC below or above 150 µg/L.

In order to elucidate if the difference in UIC can be attributed to geographical characteristics or to adverse conditions in the HR group, we removed preterm deliveries and compared only the UIC of mothers who gave birth babies with normal weight (percentile 10–90th), there was no significant difference in UIC between LR and HR groups (154.99 ± 99.37 µg/L vs. 124.64 ± 78.24; *p* = 0.077). When only women <30 years of age who took supplements (favorable clinical profile) were selected, the UIC did not show significant difference between LR and HR groups (137.97 ± 104.73 µg/L vs. 114.36 ± 70.89; *p* = 0.281).

When we compared the women who took iodine supplements to those who did not, there was no difference in maternal age, but significant differences were found in all neonatal parameters (Table 3).

In fact, the non-consumption of iodine supplements was significantly and independently associated to preterm birth (OR = 3.78, 95% CI = 1.32–10.79) (*p* = 0.010), and small for gestational age (OR = 2.87, 95% CI = 1.45–5.69) (*p* = 0.002). However, the association between UIC <150 µg/L and preterm birth did not reach statistical significance (OR = 2.86, 95% CI = 0.81–10.10) (*p* = 0.143), neither between UIC <150 µg/L and small for gestational age (OR = 1.70, 95% CI = 0.91–3.16) (*p* = 0.113).

On the other hand, amniotic fluid samples were obtained from 193 pregnant women of the low-risk group and 102 women with high-risk pregnancy. The iodine concentration in amniotic fluid was significantly higher in high-risk pregnancies compared with low-risk pregnancies (Table 2). There was a significant correlation between iodine in amniotic fluid and UIC in low-risk pregnancies (*r* = 0.195, *p* = 0.009) that disappeared in high-risk pregnancies (*r* = −0.044, *p* = 0.676). Iodine in amniotic fluid did not correlate with maternal age nor gestational age, but correlated negatively with birth weight (*r* = −0.192, *p* = 0.001) and with weight percentile (*r* = −0.174, *p* = 0.003). When we split the sample in LR and HR pregnancies, these correlations disappeared. In the group of newborns below the 10th percentile, iodine in amniotic fluid correlated with birth weight (*r* = −0.279, *p* = 0.026) and with weight percentile (*r* = −0.272, *p* = 0.029); and in those babies born at term, iodine in amniotic fluid correlated with birth weight (*r* = −0.154, *p* = 0.011) and with weight percentile (*r* = −0.154, *p* = 0.012).

### 3.3. Thyroid Function Parameters in Maternal and Cord Blood

The maternal and neonatal values of thyroid function parameters were significantly different in high-risk and low-risk pregnancies (Table 2). All parameters were higher in the low-risk group; except maternal free thyroxine (FT4) whose levels were similar in both groups. Moreover, the levels of both hormones are significantly higher in the umbilical cord compared with maternal serum: TSH (10.89 ± 7.61 mIU/L vs. 3.15 ± 2.27 mIU/L, respectively; *p* < 0.001), FT4 (16.92 ± 2.59 pmol/L vs. 12.94 ± 2.88 pmol/L, respectively; *p* < 0.001). These differences were present in both groups (LR and HR). There were positive correlations between maternal and neonatal thyroid hormones: TSH (*r* = 0.137, *p* = 0.049), and FT4 (*r* = 0.335, *p* ≤ 0.001). We did not find differences in maternal nor neonatal thyroid hormone levels according to consumption of iodine supplements, sex of the newborn, or preterm birth (Appendix A).

Whereas iodine in amniotic fluid did not correlate with maternal nor neonatal thyroid parameters, UIC significantly correlated with maternal TSH (*r* = 0.122, *p* = 0.044), neonatal TSH (*r* = 0.234, *p* < 0.001), and neonatal FT4 (*r* = 0.189, *p* = 0.005). When we considered HR and LR separately, the correlations between UIC and thyroid hormones only remained significant in the LR group (data not shown).

A negative correlation between maternal FT4 and gestational age was also found (*r* = −0.222, *p* = 0.001), which remains in both HR pregnancies (*r* = −0.350, *p* = 0.050) as well as in LR pregnancies (*r* = −0.149, *p* = 0.039). Maternal FT4 also correlated negatively with birth weight (*r* = −0.150, *p* = 0.036) in the LR group, but did not in the HR group. There was no correlation between maternal FT4 and weight percentile in LR nor HR group. A significant correlation was found between neonatal FT4 and birth weight (*r* = 0.223, *p* = 0.001), but it was not present when the participants were separated into the LR and HR groups. When birth weight was considered as a categorical variable (adjusted by gestational age and intake of supplements), it can be noticed that neonatal FT4 and iodine in amniotic fluid showed opposite behaviors (Figure 1). The ANOVA test offered significant differences according to birth weight categories for neonatal FT4 (*p* = 0.002) and for iodine in amniotic fluid (*p* = 0.019).

### 3.4. Conditions at the Time of Birth

Maternal TSH, maternal FT4, and neonatal FT4 showed significant differences according to the onset of labor (Table 4).

However, these differences can be explained by the gestational age at birth, which was substantially different among the three options: 260.65 ± 18.25 days for elective caesareans, 272.09 ± 9.98 days for spontaneous deliveries, and 275.22 ± 8.44 days for induced labor. (ANOVA test, *p* < 0.001). No differences were found for UIC or iodine in amniotic fluid depending on the onset of labor.

When preterm deliveries were excluded, maternal FT4 was significantly higher in the LR group (13.69 ± 2.37 pmol/L vs. 10.96 ± 3.24 pmol/L, respectively; *p* = 0.033) in women who had not taken supplements, but not in women who took them. Besides, if elective cesareans were excluded, maternal FT4 was significantly higher in the LR group (13.46 ± 2.36 pmol/L vs. 9.82 ± 3.64 pmol/L, respectively; *p* = 0.024) in women who took supplements, but there was not significant difference in maternal FT4 between LR and HR groups in women who did not.

We also compared maternal and neonatal parameters, according to the mode of delivery: eutocic deliveries (those that develop without the need of a doctor’s intervention), instrumental deliveries, and cesarean section (including elective as well as emergency cesarean sections). From this comparison, two variables showed significant differences: neonatal FT4 and iodine in amniotic fluid (Table 5).

Finally, linear regression models for neonatal FT4 and for iodine in amniotic fluid were developed. Maternal FT4 (beta coefficient 0.277; *p* < 0.001), mode of delivery (−0.227; *p* = 0.001), weight percentile (0.220; *p* = 0.006), and UIC (0.158; *p* = 0.026) were independently associated to neonatal FT4 levels (adjusted regression coefficient squared, *R*^2^ = 0.241; *p* < 0.001). Iodine in amniotic fluid is associated to birth weight (beta coefficient −0.152; *p* = 0.009) and mode of delivery (0.140; *p* = 0.010) (*R*^2^ = 0.114; *p* < 0.001).

## 4. Discussion

In this study, we investigated the relationship between iodine status, maternal and neonatal thyroid parameters, and birth conditions through the comparison of high-risk and low-risk pregnancies. The main finding is that neonatal FT4 levels at birth are significantly higher in low-risk pregnancies, at term, with normal birth weight; whereas iodine in amniotic fluid is lower in such circumstances. This profile, which combines high neonatal FT4 and low iodine in amniotic fluid, is also present when labor is initiated spontaneously and ends as an eutocic delivery. To our knowledge, this is the first study to demonstrate that thyroid function parameters as well as iodine metabolism can be modulated by the clinical conditions at the time of birth, particularly in high-risk pregnancies.

It is well established that TH are necessary ingredients for both the pre and postnatal developmental processes of the mammalian brain, and the amount of maternal T4 received by the fetus is directly proportional to TH action in its brain [18]. Additionally, the developing brain is a nutritionally-demanding tissue and particularly vulnerable to broad environmental fluctuations, such as specific nutrients and growth factors, that can impact long-term programming [19]. An increasing body of evidence suggests a “fetal programming” effect, which may drive offspring of women with thyroid dysfunction and/or iodine deficient to be susceptible to later onset of neurodevelopmental disorders [20,21]. But, meanwhile the early stages of pregnancy have received special attention, the epigenetic impact of perinatal conditions has scarcely been investigated.

In our study, the comparison between HR and LR groups offered two different scenarios of maternal-fetal transfer of TH and iodine metabolism. With exception of the maternal FT4 (similar in both groups), THs exhibited higher levels in the LR group. Additionally, there was an inverse relationship between UIC and iodine in amniotic fluid: the LR pregnancies presented higher UIC and lower iodine in amniotic fluid in comparison with HR pregnancies. The higher values of UIC (and major consumption of iodine supplements) in the LR group might justify the higher levels of TH [22]. In fact, the positive correlations found between UIC and these thyroid parameters would support this explanation. However, in pathological/abnormal conditions (HR group), the passage of TH as well as the relationship between iodine status and thyroid function is compromised [9,23].

Even after the onset of fetal thyroid secretion, human data show that maternal transfer still represents about 30–60% of fetal serum T4 and continues to have an important protective role in fetal neurodevelopment until birth [24]. Our results attempt to demonstrate how the maternal–fetal communication network for TH in the late stages of pregnancy should be taken into account for decision making in cases of adverse antenatal or perinatal circumstances.

Although the regulatory mechanisms of thyroid function at the time of birth are not fully understood, the positive TSH-FT4 correlation in maternal and cord blood have been previously reported [16,25]. Particularly in cord blood, it is maximal at the time of term delivery, being lower in preterm deliveries [25] and becoming negative as the days pass after birth [26]. These findings may be interpreted as stimulatory effect of TSH on FT4 secretion at the end of gestation, in order to boost the thyroid function of the newborn [27].

The relationship between maternal and neonatal thyroid parameters at the time of birth as well as their impact on fetal growth have been previously studied by others [28,29]. Our results corroborated most of their findings, such as: (i) the correlations between maternal and neonatal THS and FT4 levels, particularly in the case of FT4, which became stronger in pregnancies at term; (ii) the lower neonatal FT4 levels in children with lower birth weight (iii) the inverse association between maternal FT4 and birth weight, which is opposite to neonatal FT4. All these findings suggest that thyroid hormones may have a substantial role in regulating fetal growth and, particularly, the late stages of pregnancy, when the fetus significantly increases in size, is an important window of susceptibility to narrow variations in maternal and neonatal thyroid parameters.

Shields et al. [28] inferred that TH may influence fetal growth indirectly by affecting placental growth, even though the negative association between maternal FT4 and birth weight was unexpected and difficult to explain by the authors. This negative association has also been demonstrated in repeated measures analysis [30,31]. It might be hypothesized that maternal and neonatal FT4 represent both sides of transplacental transfer of TH: the more compromised this transfer is, the higher FT4 at maternal side and the lower at neonatal side. Our study showed that those newborns below the 10th percentile, which can be interpreted as a proxy of placental insufficiency, had significantly lower neonatal FT4 levels.

Concerning maternal FT4, it is important to highlight that their levels were similar for most of the comparisons (high/low risk, preterm/term, low/normal weight), except the onset of labor, reaching the highest levels in elective caesarean and the lowest in case of induced labor. This finding, along with the inverse correlation between maternal FT4 and gestational age, support the idea that the maternal–fetal transfer of FT4 may be particularly intensified at the time of birth, being conditioned by the timing of delivery [32]. According to this theoretical assumption, this active passage of TH at the time of birth will be optimal in spontaneous deliveries, suboptimal in case of induction and abruptly intercepted in elective caesarean. Moreover, when preterm deliveries or elective caesareans were excluded, maternal FT4 was significantly higher in the LR group, but the difference was not present in women who took supplements. Our findings suggest a lesser availability of maternal Ft4 in women with HR pregnancies that might be mitigated by the intake of supplements.

Furthermore, when the mode of delivery is considered (eutocic, instrumental, or caesarean section), we found again higher neonatal FT4 levels in eutocic deliveries in comparison with instrumental deliveries or caesarean sections. In fact, the mode of delivery appears as an influencing factor (in addition to maternal FT4 and birth percentile) on neonatal FT4 in the regression model, independently of gestational age or birth weight. Although other studies found higher levels of neonatal FT4 in vaginal deliveries in comparison with caesarean sections, the difference did not reach statistical significance [33] or was influenced by respiratory distress or other comorbidities [34]. However, it is important to remark that, in our study, we made a distinction between instrumental and eutocic vaginal deliveries, which might explain the significant differences in neonatal FT4 levels.

On the other hand, nutritional intake during pregnancy is particularly critical for directing normal fetal growth and development [35], so the influence of maternal iodine status on birth weight has been extensively studied. However, while some studies have demonstrated that babies born to women with inadequate dietary iodine intake in the third trimester had lower birth weights than those born to women with adequate intake [36,37], most of them showed no evidence of an association between UIC and birth weight [38,39,40]. This mismatch is based on a great heterogeneity among studies due to a different degree of iodine deficiency (severe vs. moderate-to-mild deficiency); different timing (first trimester, half-gestation, at birth), and different methodology in the evaluation [12,14].

In our study design, we included 3 different variables to estimate the nutritional iodine status in pregnant women: the consumption of iodine supplements, UIC, and iodine in amniotic fluid. The difference in UIC might be initially attributed to geographical characteristics of the two cities (Málaga and Granada, Spain); however, a national survey conducted by our group showed similar iodine status in these locations [41]. Additionally, when the higher risk cases were removed (by excluding preterm delivery and intrauterine growth restriction or by selecting young women with supplementation), the difference in UIC was not statistically significant.

The lower levels of UIC in HR pregnancies, may easily be explained by a lower consumption of supplements, but those pregnant women who did not take iodine supplements also showed lower values in neonatal parameters (gestational age, birth weight, and weight percentile). The fact that the risk of prematurity or small for gestational age was associated to non-consumption of iodine supplements rather than to a UIC <150 µg/L can be understood if we take into account that UIC only reflects recent intake of iodine, whereas the intake of supplements may indicate replete iodine stores [42]. Another plausible explanation is that women with low adherence to supplementation are more likely to have pregnancies at high risk for preterm delivery and/or intrauterine growth restriction, due to an overlap of underlying unfavorable factors (poor socioeconomic status, significant barriers to health services, and others) [43]. Unfortunately, our design did not allow us to establish whether the differences found in neonatal parameters in babies of the HR group would be associated with an adverse developmental environment or with the intake of supplements/iodine status.

But the most interesting contribution of our work is the study of iodine concentration in amniotic fluid. So far, we have measured iodine in amniotic fluid at mid-gestation [15] and at the time of birth in healthy pregnant women [16]: in both circumstances it showed stable levels, independently of the dietary iodine intake as well as a positive correlation with UIC. Nevertheless, the comparison of LR and HR pregnancies has shown how the levels of iodine in amniotic fluid are closely associated to birth weight and weight percentile, being significantly higher in small fetuses and also in elective caesareans.

It is known that amniotic fluid is a complex milieu which contains nutrients and growth factors that facilitate fetal growth, and there is a rapid bi-directional diffusion between the fetus and the amniotic fluid across the not-yet-keratinized fetal skin [44]. Moreover, animal studies have shown that adaptive and/or defending mechanisms against effects of iodine deficiency are present in the fetus, such as the iodide uptake through an increased expression of Na-I symporter (NIS) [45]. So it may be deduced that the fetus can uptake iodine from amniotic fluid to synthetize its own TH. According to this, a plausible explanation to our results is that this uptake of iodine would be restricted in small fetuses and suddenly interrupted in elective caesareans.

Our study, as a cross-sectional study, is limited to establish cause and effect, so the correlation seen in our data could be interpreted just in the theoretical way. However, our findings suggest that the late stages of pregnancy, when the fetus significantly increases in size, is an important window of susceptibility to even narrow thyroid function deviations.

According to our results, we can hypothesize that the availability of FT4 in cord blood is a sum of the effects produced by the maternal–fetal T4 transfer (hampered by placental resistance) and the synthesis de novo by the fetus (limited in small fetuses).

## 5. Conclusions

In summary, we can conclude that maternal and neonatal thyroid function and iodine metabolism at the time of birth is modulated by consecutive conditions: in utero environment (iodine status, placental function, amniotic fluid), the onset of labor (gestational age at birth, elective termination), and the mode of delivery.

## Figures and Tables

**Figure 1 jcm-09-00177-f001:**
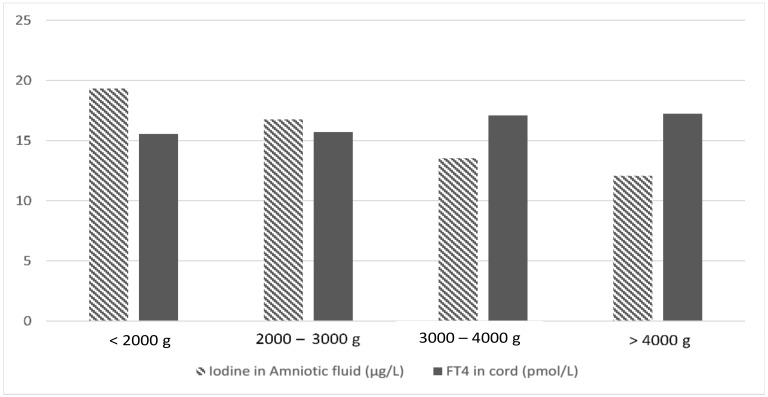
Neonatal free-thyroxine (FT4) levels and iodine in amniotic fluid, according to birthweight categories. As the birth weight increases, the levels of iodine in amniotic fluid (striped bars) decrease, whereas the concentration of FT4 in the umbilical cord (grey bars) increases.

**Table 1 jcm-09-00177-t001:** Comparison of maternal and neonatal variables between low-risk vs. high-risk groups.

	Low-Risk Group (*n* = 233)	High-Risk Group (*n* = 108)	*p*-Value
Maternal age (years)	29.78 ± 5.26	31.05 ± 5.73	0.047 *
Use of iodine supplements			
No	26 (11.2%)	31 (28.7%)	<0.001 **
Yes	207 (88.8%)	77 (71.3%)	
Onset of labor			
Spontaneous	187 (80.3%)	17 (15.7%)	
Induced	46 (19.7%)	66 (61.1%)	<0.001 **
Elective caesarean	0 (0.0%)	25 (23.2%)	
Mode of delivery			
Eutocic	176 (75.5%)	26 (24.1%)	
Instrumental	37 (15.9%)	42 (38.9%)	<0.001 **
Cesarean section	20 (8.6%)	40 (37.0%)	
Gestational age (days)	274.80 ± 8.65	270.69 ± 12.47	0.003 *
Preterm (<270 days or 37 wks)			
No	228 (97.9%)	93 (86.1%)	<0.001 **
Yes	5 (2.1%)	15 (13.9%)	
Birth weight (gr)	3358.81 ± 451.60	2754.67 ± 644.26	<0.001 *
Weight in percentiles			
Below P_10_	15 (6.4%)	55 (50.9%)	
P_10_–P_90_	180 (77.3%)	49 (45.4%)	<0.001 **
Above P_90_	38 (16.3%)	4 (3.7%)	
Sex			
Male	109 (46.8%)	65 (60.2%)	0.025 **
Female	124 (53.2%)	43 (39.8%)	
1-min Apgar score			
<5	2 (0.9%)	4 (3.7%)	0.083 **
≥5	231 (99.1%)	104 (96.3%)	
5-min Apgar score			
<7	1 (0.4%)	2 (1.9%)	0.238 **
≥7	232 (99.6%)	106 (98.1%)	
pH in umbilical cord			
arteria	7.26 ± 0.08	7.27 ± 0.07	0.600 *
venous	7.30 ± 0.07	7.30 ± 0.07	0.483 *

* Mean difference. ** Chi-squared test.

**Table 2 jcm-09-00177-t002:** Urinary iodine concentration (UIC), amniotic fluid iodine, and thyroid function parameters in mothers and newborns.

	Low-Risk Group (*n* = 233)	High-Risk Group (*n* = 108)	*p*-Value
Urinary iodine concentration (µg/L)	147.91 ± 99.07	116.49 ± 71.31	0.002
Iodine in amniotic fluid (µg/L)	12.85 ± 6.90	18.62 ± 13.23	<0.001
Maternal TSH (mIU/L)	3.31 ± 2.22	2.70 ± 1.96	0.025
Maternal FT4 (pmol/L)	13.18 ± 2.16	12.85 ± 6.51	0.078
Neonatal TSH (mIU/L)	12.20 ± 9.17	8.48 ± 4.79	<0.001
Neonatal FT4 (pmol/L)	17.26 ± 2.14	15.24 ± 3.12	<0.001

**Table 3 jcm-09-00177-t003:** Comparison according to the intake of iodine supplements.

	Iodine Supplementation (*n* = 284)	No Iodine Supplementation (*n* = 57)	*p*-Value
Maternal age (years)	30.12 ± 5.38	30.58 ± 5.95	0.599
Gestational age (days)	274.22 ± 9.75	269.29 ± 11.48	0.002 *
Birth weight (gr)	3213.78 ± 564.38	2839.78 ± 677.35	<0.001 *
Weight percentile (mean)	48.58 ± 32.77	32.18 ± 32.05	0.002
Weight in			
percentiles	51 (18.1%)	18 (40%)	
Below P_10_	195 (69.1%)	24 (53.3%)	0.003 **
P_10_–P_90_	36 (12.8%)	3 (6.7%)	
Above P_90_			

* Mean difference. ** Chi-squared test.

**Table 4 jcm-09-00177-t004:** Maternal and neonatal thyroid function parameters according to onset of labor.

	Elective Cesarean	Spontaneous Labor	Induced Labor	*p*-Value
Maternal TSH (mIU/L)	3.16 ± 1.98	3.37 ± 2.42	2.64 ± 1.58	0.033
Maternal FT4 (pmol/L)	16.24 ± 8.23	13.29 ± 2.30	12.55 ± 4.42	0.045
Neonatal TSH (mIU/L)	8.32 ± 3.23	11.73 ± 8.97	9.51 ± 6.38	0.070
Neonatal FT4 (pmol/L)	15.36 ± 1.84	17.16 ± 2.36	15.78 ± 3.10	<0.001

**Table 5 jcm-09-00177-t005:** Maternal and neonatal thyroid function parameters according to mode of delivery.

	Eutocic Delivery	Instrumental Delivery	Caesarean Section	*p*-Value
Neonatal FT4 (pmol/L)	17.29 ± 2.26	15.71 ± 3.17	15.50 ± 2.51	<0.001
Iodine in amniotic fluid (µg/L)	12.58 ± 6.89	19.82 ± 14.37	15.52 ± 9.64	<0.001
Birth weight (grams)	3292.42 ± 458.74	3034.74 ± 617.53	2900.86 ± 818.74	<0.001
Weight percentile	54.67 ± 29.55	38.51 ± 34.56	31.53 ± 35.72	<0.001

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
