# Peer review of "Iodine Status, Thyroid Function, and Birthweight: A Complex Relationship in High-Risk Pregnancies"

_jcm, 2020, doi:10.3390/jcm9010177_

Round 1

Reviewer 1 Report

The purpose of the present study was to increase the knowledge related to iodine status in pregnancies at risk and to evaluate the possible implications of iodine deficiency with fetal growth. 

The paper is certainlt interesting and present some data worthy for further studies. In particular, the inverse relationship between the levels of iodine in the amniotic fluid and the hypothesis that in conditions of high risk the fetal thyroid function is compromised is intriguing.

However, there are some critical elements. In my opinion, as the high and low risk populations come from two different Centers with distinct geographical charateristics ,  it would be preferable to have the ioduria data of a reference sample of women with low risk pregnancy from the same region to validate the comparison with the assumed population of the LR group.
Moreover , how do the authors explain that fetal growth parameters are different among  women who take or not take iodine supplementation with the fact that there is no correlation with the same parameters and maternal ioduria?Furthermore, it is not clear to me how there are no differences between LR and HR in terms of maternal Ft4 with in pregnancy is a direct expression of iodine intake. 
The authors speculate that maternal and neonatal Ft4 represent both sides of  tranplacental tranfer of thyroi hormon and  that the  greater is the placental damage and the lower  is the neonatal Ft4 with specular increase in maternal Ft4.  In this case,  if maternal Ft4 and  UIC  are not so indicative, how can we judge an adequate iodine intake in pregnancy? Wich is according to the authors the most appropriate period of gestation to verify th adequate maternal iodine intake?

Author Response

The purpose of the present study was to increase the knowledge related to iodine status in pregnancies at risk and to evaluate the possible implications of iodine deficiency with fetal growth. 

The paper is certainly interesting and present some data worthy for further studies. In particular, the inverse relationship between the levels of iodine in the amniotic fluid and the hypothesis that in conditions of high risk the fetal thyroid function is compromised is intriguing.

However, there are some critical elements. In my opinion, as the high and low risk populations come from two different Centers with distintict geographical characteristics, it would be preferable to have the ioduria data of a reference sample of women with low risk pregnancy from the same region to validate the comparison with the assumed population of the LR group.

Dear reviewer,

Thank you very much for your comments. Our group had previously conducted a national survey regarding iodine status of a representative sample of adult population in Spain (Soriguer et al. Iodine intake in the adult [email protected] study. Clinical Nutrition 2012; 31:882-8, reference 42 in the manuscript), where we found comparable data between these two locations.

Within this study, when we excluded preterm babies and compared only those women who delivered babies with normal birth weight (percentiles 10-90th), there were no significant differences in UIC according to LR or HR groups. Additionally, when we selected only women<30 years of age who were taken iodine supplements to be compared between LR and HR group, the UIC was rather similar. So, our findings might be explained by differences in maternal and neonatal parameters associated with a high risk profile. We have added this information at the results section (Lines 178-184) and included a paragraph at the discussion (Lines 348-353).

Moreover, how do the authors explain that fetal growth parameters are different among women who take or not take iodine supplementation with the fact that there is no correlation with the same parameters and maternal ioduria?

This is certainly a good point to discuss. Fetal growth parameters may be affected by many factors such as socioeconomic status, educational level, nutritional deficiencies, etc. Unfortunately, we cannot elucidate the impact of this background with our design. In any case, we have enriched the discussion with the following paragraph regarding this particular issue (Lines 360-365).

“Another plausible explanation is that women with low adherence to supplementation are more likely to have pregnancies with high risk of preterm delivery and/ or intrauterine growth restriction, due to an overlap of underlying unfavourable factors (poor socioeconomic status, low education level, significant barriers to health services, and others) [41]. Unfortunately, our design did not allow to establish whether the differences found in neonatal parameters in babies of HR group would be associated with an adverse developmental environment or with the intake of supplements/ iodine status”.

Furthermore, it is not clear to me how there are no differences between LR and HR in terms of maternal Ft4 with in pregnancy is a direct expression of iodine intake. 

As we have seen in our study, maternal Ft4 are clearly affected by factors such as gestational age or the onset of labour, being significantly higher in cases of preterm delivery as well as elective caesarean section. When we removed those conditions and compared maternal Ft4 in LR and HR groups, there were significant differences, but only for women who did not take iodine supplements. These findings suggest a lesser availability of maternal Ft4 in women with HR pregnancies that might be mitigated by the intake of supplements. We have added this information in the manuscript (Results Section Lines 244-249; Discussion Lines 324-328).

The authors speculate that maternal and neonatal Ft4 represent both sides of tranplacental tranfer of thyroi hormon and  that the  greater is the placental damage and the lower  is the neonatal Ft4 with specular increase in maternal Ft4.  In this case,  if maternal Ft4 and  UIC  are not so indicative, how can we judge an adequate iodine intake in pregnancy? Wich is according to the authors the most appropriate period of gestation to verify th adequate maternal iodine intake?

At this moment, the search for a reliable parameter to assess iodine status in individuals (not at population level) is challenging. In our opinion, the parameter which has shown an evident impact on neonatal and infant development is maternal Ft4 in the first trimester of pregnancy. According to that, the optimal moment to verify the maternal iodine status is preconceptionally or at the very early stages of pregnancy, and recommend iodine supplements when it needs to guarantee normal levels of maternal Ft4 during the first half of pregnancy.

Although the aim of our study was to demonstrate that maternal-fetal transference of thyroid hormones around delivery can be affected by obstetrics and perinatal factors, our results also showed that some unfavourable clinical conditions can be counterbalanced by a good nutritional iodine status.

Reviewer 2 Report

In the present manuscript, Velasco et al. analyzed the relationship between iodine, thyroid hormones and fetal growth in high-risk pregnancies and found that FT4 concentration was higher in cord blood from low risk group respect to high risk pregnancies, whereas iodine concentration was lower in amniotic fluid from low risk group respect to high risk group. They concluded that an adverse intrauterine environment can compromise the availability of FT4 in cord blood as well as the iodine metabolism in the fetus.

The manuscript is potentially interesting and well organized. The results make sense and the conculsion are congruent.

Author Response

In the present manuscript, Velasco et al. analyzed the relationship between iodine, thyroid hormones and fetal growth in high-risk pregnancies and found that FT4 concentration was higher in cord blood from low risk group respect to high risk pregnancies, whereas iodine concentration was lower in amniotic fluid from low risk group respect to high risk group. They concluded that an adverse intrauterine environment can compromise the availability of FT4 in cord blood as well as the iodine metabolism in the fetus.

The manuscript is potentially interesting and well organized. The results make sense and the conculsion are congruent.

Dear reviewer,

Thank you very much for appreciating our work.

Reviewer 3 Report

The article "Iodine status, Thyroid function and birthweight: a complex relationship in high risk pregnancies” is an interesting article about thyroid function in maternal and neonatal samples. This article is interesting, well organized and experiments are well designed. However I do not understand why both TSH and FT4 decrease in high-risk group, both in maternal and neonatal samples. A decrease in TSH leads to an increase of T4 levels and authors observe opposite results. From my point of view this issue should be clariffied and discussed.

Author Response

The article "Iodine status, Thyroid function and birthweight: a complex relationship in high risk pregnancies” is an interesting article about thyroid function in maternal and neonatal samples. This article is interesting, well organized and experiments are well designed. However I do not understand why both TSH and FT4 decrease in high-risk group, both in maternal and neonatal samples. A decrease in TSH leads to an increase of T4 levels and authors observe opposite results. From my point of view this issue should be clariffied and discussed.

Dear reviewer,

Thank you very much for your comments. You are absolutely right to highlight the positive TSH-FT4 correlation in maternal as well as in cord blood. We have added the following paragraph at the discussion:

“Although the regulatory mechanisms of thyroid function at the time of birth are not fully understood, the positive TSH-FT4 correlation in maternal and cord blood have been previously reported [16, 25]. Particularly in cord blood, it is maximal at the time of term delivery, being lower in preterm deliveries [25] and becoming negative as the days pass after birth [26]. These findings may be interpreted as stimulatory effect of TSH on FT4 secretion at the end of gestation, in order to boost the thyroid function of the newborn [27].”

Reviewer 4 Report

In this paper, the authors aimed to evaluate the potential differences about iodine status, maternal and neonatal thyroid parameters and birth weight between high-risk and low-risk pregnancies. The manuscript is generally well written, and provides important insights in the effects of thyroid function and iodine status on fetal growth.

The comments thereinafter are in order of appearance.

Methods.

Lines 83-85. Consider rephrasing this sentence for clarity, as  the definition of low-risk pregnant women is confusing.

Line 198. Consider deleting the second “in Malaga”.

Line 115. Please give the full name of CV.

Results.

This section contains numerous findings. I believe that at least one more table would facilitate understanding, in particular results reported at lines 142-161, 173-198 should be included in a specific table.

Table 2. Are the values of urinary iodine concentration mean values? They are different from those reported in the text.

Lines 173-173. The nature of these associations is unclear. Is the first one relative to UIC < 150 μg/L and the second one relative to UIC > 150 μg/L?

Line 191. Please correct the upper case in “iodine”.

Lines 195-198. What is the nature of these inverse correlations? Please rephrase for better clarity.

Line 202. Consider changing “which” to “whose”.

Lines 240-247. These results should be included in Table 3.

Discussion

Line 326. Please provide the full name of ID.

Author Response

In this paper, the authors aimed to evaluate the potential differences about iodine status, maternal and neonatal thyroid parameters and birth weight between high-risk and low-risk pregnancies. The manuscript is generally well written, and provides important insights in the effects of thyroid function and iodine status on fetal growth.

The comments thereinafter are in order of appearance.

Dear Reviewer,

Thank you very much for helping us to improve the manuscript.

Methods.

Lines 83-85. Consider rephrasing this sentence for clarity, as the definition of low-risk pregnant women is confusing.

We have corrected the sentence in order to make it easier to understand. Additionally, the factors to be considered as high-risk pregnancy are described in lines 87-90.

Line 198. Consider deleting the second “in Malaga”.

The second “in Málaga” has been deleted.

Line 115. Please give the full name of CV.

 We have added the full term “Coefficient of variation”

Results.

This section contains numerous findings. I believe that at least one more table would facilitate understanding, in particular results reported at lines 142-161, 173-198 should be included in a specific table.

The results previously reported at lines 180-185 are currently presented as Table 3.

Table 2. Are the values of urinary iodine concentration mean values? They are different from those reported in the text.

Table 2 contains mean± standard deviation values of UIC, whereas median values are reported in the text.

Lines 173-173. The nature of these associations is unclear. Is the first one relative to UIC < 150 μg/L and the second one relative to UIC > 150 μg/L?

We have clarified these results (Lines 173-175).

Line 191. Please correct the upper case in “iodine”.

It has been corrected.

Lines 195-198. What is the nature of these inverse correlations? Please rephrase for better clarity.

These results have been rephrased. Lines 203-205.

Line 202. Consider changing “which” to “whose”

The change is done.

Lines 240-247. These results should be included in Table 3.

Table 3 (currently table 4) contains parameters according to the onset of labor (elective caesarean, spontaneous labor, induced labor). The results described in lines 240-247 were obtained according to the mode of delivery (eutocic deliveries, instrumental deliveries and caesarean section). Since the onset of labor and the mode of delivery are not equivalent concepts, we have included these results in a new table (Table 5).

Discussion

Line 326. Please provide the full name of ID.

ID has been replaced by iodine deficiency.

Round 2

Reviewer 1 Report

The answer to the objections and the relative clarifications in the text are sufficient for me to resolve the reservation.